

# A simple method for data partitioning based on relative evolutionary rates

Jadranka Rota[1], Tobias Malm[2], Nicolas Chazot[1], Carlos Peña[3] and Niklas Wahlberg[1]

[1] Department of Biology, Lund University, Lund, Sweden
[2] Department of Zoology, Swedish Museum of Natural History, Stockholm, Sweden
[3] HipLead, San Francisco, CA, United States of America

## ABSTRACT

**Background**. Multiple studies have demonstrated that partitioning of molecular datasets is important in model-based phylogenetic analyses. Commonly, partitioning is done *a priori* based on some known properties of sequence evolution, e.g. differences in rate of evolution among codon positions of a protein-coding gene. Here we propose a new method for data partitioning based on relative evolutionary rates of the sites in the alignment of the dataset being analysed. The rates are inferred using the previously published Tree Independent Generation of Evolutionary Rates (TIGER), and the partitioning is conducted using our novel python script RatePartitions. We conducted simulations to assess the performance of our new method, and we applied it to eight published multi-locus phylogenetic datasets, representing different taxonomic ranks within the insect order Lepidoptera (butterflies and moths) and one phylogenomic dataset, which included ultra-conserved elements as well as introns.

**Methods**. We used TIGER-rates to generate relative evolutionary rates for all sites in the alignments. Then, using RatePartitions, we partitioned the data into partitions based on their relative evolutionary rate. RatePartitions applies a simple formula that ensures a distribution of sites into partitions following the distribution of rates of the characters from the full dataset. This ensures that the invariable sites are placed in a partition with slowly evolving sites, avoiding the pitfalls of previously used methods, such as *k*-means. Different partitioning strategies were evaluated using BIC scores as calculated by PartitionFinder.

**Results**. Simulations did not highlight any misbehaviour of our partitioning approach, even under difficult parameter conditions or missing data. In all eight phylogenetic datasets, partitioning using TIGER-rates and RatePartitions was significantly better as measured by the BIC scores than other partitioning strategies, such as the commonly used partitioning by gene and codon position. We compared the resulting topologies and node support for these eight datasets as well as for the phylogenomic dataset.

**Discussion**. We developed a new method of partitioning phylogenetic datasets without using any prior knowledge (e.g. DNA sequence evolution). This method is entirely based on the properties of the data being analysed and can be applied to DNA sequences (protein-coding, introns, ultra-conserved elements), protein sequences, as well as morphological characters. A likely explanation for why our method performs better than other tested partitioning strategies is that it accounts for the heterogeneity in the data to a much greater extent than when data are simply subdivided based on prior knowledge.

Corresponding author
Jadranka Rota, jadranka.rota@biol.lu.se

## INTRODUCTION

Phylogenetic analysis of DNA sequences is based on models of molecular evolution that estimate parameters such as base frequencies, substitution rates among nucleotides, as well as among-site rate variation. To reduce the heterogeneity in the data, datasets are often partitioned into subsets that are deemed to have undergone more similar molecular evolution. A number of studies have demonstrated that partitioning of data is important (*Nylander et al., 2004*; *Brandley, Schmitz & Reeder, 2005*; *Brown & Lemmon, 2007*; *Rota, 2011*; *Rota & Wahlberg, 2012*; *Kainer & Lanfear, 2015*), especially for model-based phylogenetic analyses, which are known to be more sensitive to underparameterization than overparameterization (*Huelsenbeck & Rannala, 2004*; *Lemmon & Moriarty, 2004*; *Nylander et al., 2004*).

Today, in most phylogenetic studies, partitions are defined *a priori* by the user, commonly by gene, gene and codon position, stems vs. loops in ribosomal RNA, or another feature of the sequence that the user believes to be important. In several studies, partitioning of protein-coding genes by gene and codon position was demonstrated to be a better option when compared to not partitioning or partitioning by gene (*Nylander et al., 2004*; *Brandley, Schmitz & Reeder, 2005*; *Brown & Lemmon, 2007*; *Miller, Bergsten & Whiting, 2009*; *Rota, 2011*). This approach is practical when a dataset consists of only a few genes. However, when data come from tens (or hundreds) of genes, this approach becomes unwieldy, although there are methods that allow one to combine many *a priori* established partitions into fewer, based on model testing with programs such as PartitionFinder (*Lanfear et al., 2012*).

Using a method described by *Cummins & McInerney (2011)*, it is possible to partition a dataset in a more objective way, based on the properties of the data. The method takes into account the relative evolutionary rates of characters by comparing the patterns in character-state distributions in homologous characters (i.e., nucleotides or amino acids in a molecular alignment or characters in a morphological matrix). Each character thus receives a value for its evolutionary rate, which is based on comparisons to all other characters in the matrix. The rate values can then be used to group characters with similar rates by dividing the range of rates into bins, which can be user-defined so as to span equal ranges of rates. This usually leads to the first bin containing characters that are invariable, and the last bin consisting of characters with the highest relative rate of change (*Cummins & McInerney, 2011*). This method was first implemented in the program TIGER–Tree Independent Generation of Evolutionary Rates (*Cummins & McInerney, 2011*), and later on in a much faster program TIGER-rates (*Frandsen et al., 2015*).

Originally, the method was developed to identify and exclude the fastest-evolving characters in a dataset, but this approach has potential problems (see *Simmons & Gatesy, 2016*). We have extended the TIGER method to partitioning the data by sorting characters

into data subsets with similar relative rates of evolution (*Rota & Wahlberg, 2012*; *Rota & Miller, 2013*; *Wahlberg et al., 2014*), where we arbitrarily combined neighbouring TIGER bins to form data partitions with enough characters for analysis as TIGER bins can contain too few characters for reliable parameter estimation. A similar approach has been used in a number of studies (*Kaila et al., 2013*; *Rota & Miller, 2013*; *Heikkilä et al., 2014*; *Matos-Maravi et al., 2014*; *Wahlberg et al., 2014*; *Edger et al., 2015*; *Kristensen et al., 2015*; *Rajaei et al., 2015*; *Ounap, Viidalepp & Truuverk, 2016*), and although this method works quite well, the downside is that it requires the user to make a subjective decision about the final partitioning strategy.

Recently, a different way of using TIGER together with *k*-means was described by *Frandsen et al. (2015)*. They compared their new method to traditional *a priori* defined partitions, as well as to site rates calculated using a maximum likelihood function. In all test cases, partitioning by both TIGER calculated rates and likelihood calculated rates performed better than traditional methods, with likelihood rates doing much better (*Frandsen et al., 2015*). However, the *k*-means algorithm has been found to place all invariable characters into one partition (*Baca et al., 2017*), which causes a problem because this partition then includes characters that *can* vary given enough time, but *do not* in the dataset at hand. A very high likelihood due to lack of variation in the data results in selection of a wrong model. Indeed, the *k*-means algorithm has now been disabled for molecular data in PartitionFinder2 (https://github.com/brettc/partitionfinder/commit/19d7fe41d2e469c131a5b0cc30184a069867b7f2 accessed 13 November 2017).

Here, we describe a simple and objective method for partitioning using TIGER-rates. TIGER-rates is used for sorting of sites based on their relative evolutionary rates, but now we introduce an algorithm—RatePartitions—for dividing the sites among partitions in an objective way. This method has already been used in several published studies (*Heikkilä et al., 2015*; *Rota, Pena & Miller, 2016*; *Rota et al., 2016*; *Sahoo et al., 2016*; *Dhungel & Wahlberg, 2018*; *Seraphim et al., 2018*). We further assess the performance of the method by conducting analyses of simulated datasets, as well as applying it to eight published phylogenetic datasets and one phylogenomic dataset including ultra-conserved elements (UCEs) and introns.

## MATERIALS & METHODS

### RatePartitions

Although it is technically incorrect to use the word 'partition' when referring to a data subset, we use 'partition' in that sense since this is commonly done in phylogenetics. When partitioning is carried out using TIGER, one must take into account the general properties of the data. One of these properties is that with standard DNA sequence data of protein-coding genes, one to two thirds of the data typically consist of invariable characters. These tend to be binned together to the exclusion of other data when using the TIGER binning strategy or the *k*-means algorithm (*Baca et al., 2017*). A partition made of only such data can lead to very high likelihood estimates for it although the assumption that all invariant sites observed in the dataset are truly invariant is most likely wrong. Thus, it

is advisable to include a number of slowly evolving characters to create a data partition with low variation. To deal with that problem we developed RatePartitions–an algorithm which works in the following way. The dataset is first run in TIGER or TIGER-rates to calculate the relative rate of evolution for each site (character). These values can range from 1 (invariable sites) to 0 (no common patterns, i.e., the fastest-evolving sites). The sites are then combined into partitions using RatePartitions, which applies a simple formula that ensures a distribution of sites into partitions following the distribution of rates of the characters from the full dataset. This leads to larger partitions for characters with slower rates and, conversely, smaller partitions for those with higher rates. Preliminary tests using MrModeltest v2.3 (*Nylander, 2004*) and PartitionFinder v.1.0.0 (*Lanfear et al., 2012*) suggested that this strategy led to models with uniform rate variation within partitions.

RatePartitions is a PYTHON script (Script S1, available at GitHub: https://github.com/jadrankarota/RatePartitions) that determines the rate-spans for a variable number of partitions based on a user-specified division factor and the range of rates calculated by TIGER-rates, and subsequently defines character sets for each partition. The rate-spans are calculated for the first (and slowest) partition with the following function:

$$z = x - ((x - y)/d)$$

and for the remaining partitions:

$$z = x - ((x - y)/(d + p * 0.3))$$

where $z$ is the lower limit of the rate-span, $x$ is the upper limit of the rate-span (determined iteratively for each partition, i.e., $z$ becomes $x$ in the following iteration), $y$ is the minimum value of rates for the entire dataset, $d$ is a user defined division factor (which must be greater than 1; a higher number gives a greater number of partitions) and $p$ is the partition number (when >1), which is multiplied by a fixed value of 0.3. The latter reduces the rate-span exponentially as partition number grows, which we found leads to partitions with more uniform rate variation for model-based analyses. Thus, for a dataset with rates ranging from 1 to 0.2 and with $d$ set to 1.5, the first partition will consist of all characters with rates between 1 and $1 - ((1 - 0.2)/1.5) = 0.4667$. For partition 2, $x = 0.4667$ and this partition will include characters with rates between 0.4667 and $0.4667 - ((0.4667 - 0.2)/(1.5 + 2*0.3) = 0.3397$, and so on until less than 10% of all characters are remaining. At this point the iterations are stopped and the remaining characters are placed into their own partition (which becomes the last and fastest-evolving partition).

## Simulations

To test the behaviour of our partitioning approach, we simulated DNA sequences on simple trees, then we re-estimated the trees using either TIGER-rates and RatePartitions or the standard partitioning approaches, and we compared the results. We simulated DNA sequences on two different topologies: one asymmetrical (outgroup, P, (O, (N, (M, (L, (K, (J, (I, (H, (G, (F, (E, (D, (C, (B, A)))))))))))))))) and one symmetrical (outgroup, (((A, B), (C, D)), ((E, F), (G, H))), (((I, J), (K, L)), ((M, N), (O, P)))). Each tree had 16 terminal branches and all terminal branches were set to a length of 0.1. Using INDELibleV1.03

(*Fletcher & Yang, 2009*), we simulated for each tree category 14 different datasets of 4,000 base pairs (bp) divided into four partitions of 1,000 bp each. The datasets varied in base composition, substitution models, amount of among-site rate variation, and internal branch length of the tree. We simulated 10 replicates with each set of simulation conditions for both the asymmetrical (AS: simulations on the asymmetrical tree) and symmetrical tree (SS: simulations on the symmetrical tree), resulting in a total of 280 simulated datasets. Full details with all the variables are presented in Table 1. Furthermore, to test for the effect of missing data, we simulated datasets with one of the four partitions randomly removed for three random taxa from the AS1, AS2, SS1, and SS2 sets of simulations (each with 10 replicates). All simulated datasets are available in a public online repository Zenodo (doi: 10.5281/zenodo.1251684), as well as Data S1 in the online version of this article.

From each of the resulting simulated datasets we re-estimated the tree using maximum likelihood implemented in RAxML 8.4 (*Stamatakis, 2014*) either by (1) partitioning the dataset into the four partitions (4-part) that were simulated, or (2) using TIGER-rates and RatePartitions (TIG). Finally, we compared the resulting trees from these analyses to the original tree used for simulating the dataset using normalized Robinson-Foulds tree distance (RFD) (*Robinson & Foulds, 1981*). The results are reported as averages across 10 replicates for each set of simulation conditions.

## Phylogenetic empirical datasets

We analysed eight previously published lepidopteran datasets (*Kodandaramaiah et al., 2010*; *Sihvonen et al., 2011*; *Penz, Devries & Wahlberg, 2012*; *Rota & Wahlberg, 2012*; *Zahiri et al., 2013*; *Matos-Maravi et al., 2014*; *Wahlberg et al., 2014*; *Rönkä et al., 2016*) (Table 2). From the published datasets we excluded sites from the alignment that had more than 80% of missing data unless the whole dataset had 1% or fewer of such sites (data were excluded from Arctiina, Geometridae, *Morpho*, and Pieridae; final amount of missing data is given in Table S1). All datasets are provided as Data S1 (and deposited in Zenodo doi: 10.5281/zenodo.1252828). The datasets varied in base pair length from 4,435 to 6,372 and in number of taxa from 31 to 164 (Table 2). All datasets included one mitochondrial gene (COI) and four to seven nuclear genes that are commonly used in lepidopteran phylogenetics (CAD, EF-1$\alpha$, GAPDH, IDH, MDH, RpS5, wingless) (*Wahlberg & Wheat, 2008*). We compared 14 partitioning strategies (Table 3), including user-defined ones such as partitioning by gene and by gene and codon position, and a number of different strategies devised based on the relative evolutionary rates assigned by TIGER-rates and division of sites into partitions using the RatePartitions algorithm. We varied the parameter $d$ in the RatePartitions algorithm between 1.5 and 4.5 in increments of 0.5. For comparison of the partitioning strategies we used the BIC score as calculated by PartitionFinder 1.1 (*Lanfear et al., 2012*). We did two types of searches with PartitionFinder. The first was a user-defined search for direct evaluation of the partitioning strategy obtained with TIGER-rates and RatePartitions. The second was a greedy search (we use "Gr" to mark which strategies included a greedy search), which searches for partitions with similar parameter estimates and combines them so as to reduce the final number of partitions. For example, for a dataset with eight genes that are *a priori* partitioned by gene and codon position (24 partitions), a
**Table 1** **Parameter combination for simulated datasets.** We simulated datasets for a set of 14 parameter combinations (S1 to S14) on asymmetrical and symmetrical trees. All datasets had four partitions, each with 1,000 base pairs. We varied base composition (percentage T, C, A, G), substitution models (F81 (*Felsenstein, 1981*); GTR (*Tavaré, 1986*) with different transition rates for different partitions shown in the table in the following order C↔T, A↔T, G↔T, A↔C, G↔C, and G↔A set to 1), proportion of invariable sites (P inv), the alpha parameter of the Gamma distribution for modelling of the among-site rate variation (α), and the internal branch length (internal). All terminal branches were set to 0.1.

| | T | C | A | G | Substitution model | P inv | | α | | Internal |
|---|---|---|---|---|---|---|---|---|---|---|
| **S1, S2** | | | | | | S1 | S2 | S1 | S2 | |
| Partition 1 | 40% | 10% | 40% | 10% | F81 | 0.00 | 0.80 | 0.50 | 0.50 | 0.01 |
| Partition 2 | 30% | 20% | 30% | 20% | F81 | 0.00 | 0.60 | 0.50 | 0.50 | 0.01 |
| Partition 3 | 20% | 30% | 20% | 30% | F81 | 0.00 | 0.40 | 0.50 | 0.50 | 0.01 |
| Partition 4 | 10% | 40% | 10% | 40% | F81 | 0.00 | 0.20 | 0.50 | 0.50 | 0.01 |
| **S3, S4** | | | | | | S3 | S4 | S3 | S4 | |
| Partition 1 | 40% | 10% | 40% | 10% | F81 | 0.00 | 0.80 | 0.50 | 0.50 | 0.05 |
| Partition 2 | 30% | 20% | 30% | 20% | F81 | 0.00 | 0.60 | 0.50 | 0.50 | 0.05 |
| Partition 3 | 20% | 30% | 20% | 30% | F81 | 0.00 | 0.40 | 0.50 | 0.50 | 0.05 |
| Partition 4 | 10% | 40% | 10% | 40% | F81 | 0.00 | 0.20 | 0.50 | 0.50 | 0.05 |
| **S5, S6** | | | | | | S5 | S6 | S5 | S6 | |
| Partition 1 | 40% | 10% | 40% | 10% | F81 | 0.00 | 0.80 | 0.50 | 0.50 | 0.001 |
| Partition 2 | 30% | 20% | 30% | 20% | F81 | 0.00 | 0.60 | 0.50 | 0.50 | 0.001 |
| Partition 3 | 20% | 30% | 20% | 30% | F81 | 0.00 | 0.40 | 0.50 | 0.50 | 0.001 |
| Partition 4 | 10% | 40% | 10% | 40% | F81 | 0.00 | 0.20 | 0.50 | 0.50 | 0.001 |
| **S7, S8** | | | | | | S7 | S8 | S7 | S8 | |
| Partition 1 | 40% | 10% | 40% | 10% | F81 | 0.00 | 0.80 | 0.50 | 0.50 | 0.005 |
| Partition 2 | 30% | 20% | 30% | 20% | F81 | 0.00 | 0.60 | 0.50 | 0.50 | 0.005 |
| Partition 3 | 20% | 30% | 20% | 30% | F81 | 0.00 | 0.40 | 0.50 | 0.50 | 0.005 |
| Partition 4 | 10% | 40% | 10% | 40% | F81 | 0.00 | 0.20 | 0.50 | 0.50 | 0.005 |
| **S9, S10** | | | | | | S9 | S10 | S9 | S10 | |
| Partition 1 | 40% | 10% | 40% | 10% | GTR 0.01 0.01 0.001 0.01 0.001 | 0.00 | 0.80 | 0.50 | 0.50 | 0.005 |
| Partition 2 | 30% | 20% | 30% | 20% | GTR 1 0.1 0.1 0.2 0.1 | 0.00 | 0.60 | 0.50 | 0.50 | 0.005 |
| Partition 3 | 20% | 30% | 20% | 30% | GTR 0.7 0.4 0.2 0.15 0.15 | 0.00 | 0.40 | 0.50 | 0.50 | 0.005 |
| Partition 4 | 10% | 40% | 10% | 40% | GTR 0.8 0.5 0.5 0.2 0.2 | 0.00 | 0.20 | 0.50 | 0.50 | 0.005 |
| **S11, S12** | | | | | | S11 | S12 | S11 | S12 | |
| Partition 1 | 40% | 10% | 40% | 10% | GTR 0.01 0.01 0.001 0.01 0.001 | 0.00 | 0.80 | 0.70 | 0.10 | 0.005 |
| Partition 2 | 30% | 20% | 30% | 20% | GTR 1 0.1 0.1 0.2 0.1 | 0.00 | 0.60 | 0.50 | 0.30 | 0.005 |
| Partition 3 | 20% | 30% | 20% | 30% | GTR 0.7 0.4 0.2 0.15 0.15 | 0.00 | 0.40 | 0.30 | 0.50 | 0.005 |
| Partition 4 | 10% | 40% | 10% | 40% | GTR 0.8 0.5 0.5 0.2 0.2 | 0.00 | 0.20 | 0.10 | 0.70 | 0.005 |
| **S13, S14** | | | | | | S13 | S14 | S13 | S14 | |
| Partition 1 | 40% | 10% | 40% | 10% | GTR 0.01 0.01 0.001 0.01 0.001 | 0.00 | 0.80 | 0.50 | 0.50 | 0.005 |
| Partition 2 | 30% | 20% | 30% | 20% | GTR 1 0.1 0.1 0.2 0.1 | 0.00 | 0.60 | 0.50 | 0.50 | 0.01 |
| Partition 3 | 20% | 30% | 20% | 30% | GTR 0.7 0.4 0.2 0.15 0.15 | 0.00 | 0.40 | 0.50 | 0.50 | 0.015 |
| Partition 4 | 10% | 40% | 10% | 40% | GTR 0.8 0.5 0.5 0.2 0.2 | 0.00 | 0.20 | 0.50 | 0.50 | 0.02 |

greedy search may result in a total of nine partitions because some of the original partitions were combined into a larger subset of data with similar parameter values. BIC was chosen as a statistical model evaluation metric because it usually selects simpler models than AICc and it has been shown to perform well in model selection for phylogenetic analysis

**Table 2** **Eight empirical phylogenetic datasets analysed.** List of analysed datasets providing the reference, the number of sampled taxa and gene regions in the dataset, the length of the dataset in base pairs (bp), and the DOI number for accessing each dataset at Zenodo.

| Taxon | Study | No. taxa | No. genes | base pairs |
|---|---|---|---|---|
| Arctiina | *Rönkä et al. (2016)* | 113 | 8 | 5,809 |
| Calisto | *Matos-Maravi et al. (2014)* | 90 | 6 | 5,297 |
| Choreutidae | *Rota & Wahlberg (2012)* | 41 | 8 | 6,293 |
| Coenonymphina | *Kodandaramaiah et al. (2010)* | 69 | 5 | 4,435 |
| Geometridae | *Sihvonen et al. (2011)* | 164 | 8 | 5,998 |
| Morpho | *Penz, Devries & Wahlberg (2012)* | 31 | 8 | 6,372 |
| Noctuidae | *Zahiri et al. (2013)* | 78 | 8 | 6,365 |
| Pieridae | *Wahlberg et al. (2014)* | 110 | 8 | 6,247 |

**Table 3** **List of partitioning strategies evaluated for each of the analysed datasets.** TIGER refers to the program that assigns each site in the alignment a relative evolutionary rate, and $d$ is the division factor in the RatePartitions script used to group sites into subsets based on their relative evolutionary rates. See text for more details.

| Partitioning strategy | Description |
|---|---|
| Gene | each gene fragment as separate subset |
| GeneGr | as above but with PF greedy algorithm combined into similar subsets |
| Codon | each codon position of each gene as separate subset |
| CodonGr | as above but with PF greedy algorithm combined into similar subsets |
| TIG1.5 | TIGER partitioning strategy with $d = 1.5$ |
| TIG2.0 | TIGER partitioning strategy with $d = 2.0$ |
| TIG2.5 | TIGER partitioning strategy with $d = 2.5$ |
| TIG3.0 | TIGER partitioning strategy with $d = 3.0$ |
| TIG3.5 | TIGER partitioning strategy with $d = 3.5$ |
| TIG3.5Gr | as above but with PF greedy algorithm combined into similar subsets |
| TIG4.0 | TIGER partitioning strategy with $d = 4.0$ |
| TIG4.0Gr | as above but with PF greedy algorithm combined into similar subsets |
| TIG4.5 | TIGER partitioning strategy with $d = 4.5$ |
| TIG4.5Gr | as above but with PF greedy algorithm combined into similar subsets |

(*Abdo et al., 2005*; *Ripplinger & Sullivan, 2008*). We refer to analyses with different values of $d$ as TIG1.5, TIG2.0, etc. The greedy search was not performed on TIG1.5, TIG2.0, TIG2.5, and TIG3.0 partitioning strategies because these were shown to have inferior BIC values in preliminary analyses.

We also conducted RAxML analyses of all the eight empirical datasets to compare the resulting trees between those from partitioning by gene and codon position (CodonGr) and the TIGER partitioning strategy with the best BIC score: TIG4.0Gr for Arctiina, Coenonymphina, Geometridae, and *Morpho*; and TIG4.5Gr for *Calisto*, Choreutidae, Noctuidae, and Pieridae. We used RFDs to quantify topological differences in resulting trees and we carried out a visual comparison of the trees to determine where the topological differences are located.

**Table 4  Tree topology comparisons for simulated datasets.** Shown are means for Robinson-Fould Distances between the inferred trees from simulated data compared with the true tree using the 4-partitioning strategy and the TIGER strategy for each of the 14 simulation conditions on both the asymmetrical (AS) and symmetrical trees (SS), as well as overall means and standard deviations across all of the AS and SS simulations for both the 4-part and the TIGER partitioning strategy.

| | AS 4-part | AS TIGER | SS 4-part | SS TIGER |
|---|---|---|---|---|
| 1 | 0 | 0 | 0 | 0 |
| 2 | 0.4 | 0.8 | 0.8 | 1.4 |
| 3 | 0 | 0 | 0 | 0 |
| 4 | 0 | 0 | 0 | 0 |
| 5 | 10.2 | 12.8 | 15 | 19.6 |
| 6 | 14.8 | 21.2 | 24.6 | 26.6 |
| 7 | 0.6 | 1.4 | 0.6 | 1.6 |
| 8 | 2.4 | 4.2 | 5.8 | 7 |
| 9 | 0.2 | 0.6 | 2 | 3.6 |
| 10 | 2.8 | 3.6 | 1.8 | 5.2 |
| 11 | 1.2 | 2.6 | 3.2 | 5.6 |
| 12 | 6.4 | 7 | 6.2 | 8.8 |
| 13 | 0 | 0 | 0 | 0.2 |
| 14 | 0 | 0.2 | 0 | 0 |
| mean | 2.8 | 3.9 | 4.3 | 5.7 |
| S.D. | 4.6 | 6.1 | 7.1 | 8.0 |

## Phylogenomic empirical dataset

To test the performance of our partitioning approach on a phylogenomic dataset, we analysed a dataset that includes *ca.* 1,500 UCEs, nuclear introns from 15 loci, and three mitochondrial gene regions, totalling over 600 kilobases for 18 taxa of gallopheasants and is a difficult dataset because of a presumed rapid evolutionary radiation that occurred in this group (*Meiklejohn et al., 2016*). Using TIGER-rates and RatePartitions with $d = 10$, 15, and 20 and running a greedy search in PartitionFinder, we identified the best partitioning strategy as TIG15. Finally, we analysed the dataset using IQ-TREE v.1.6.0 (*Nguyen et al., 2015*; *Chernomor, Von Haeseler & Minh, 2016*) with 1000 ultrafast bootstrap replicates (*Hoang et al., 2018*) either by partitioning according to TIGER-rates and RatePartitions or following *Meiklejohn et al. (2016)* partitioning strategy.

## RESULTS

### Simulations

A comparison of trees resulting from analyses of the simulated datasets divided into four partitions (four-part; replicating simulation conditions) shows that they performed slightly better than those with trees resulting from TIGER-rates and RatePartition analyses (TIG) as measured with RFD: across all the simulations on asymmetrical trees (AS) the RFDs for the four-part analyses have a mean of $2.8 \pm 4.6$, while TIG analyses have a mean of $3.9 \pm 6.2$; and across all the simulations on symmetrical trees (SS) for the four-part the mean was $4.3 \pm 7.1$, while for TIG it was $5.7 \pm 8.1$ (Table 4, Fig. 1). In four datasets, for

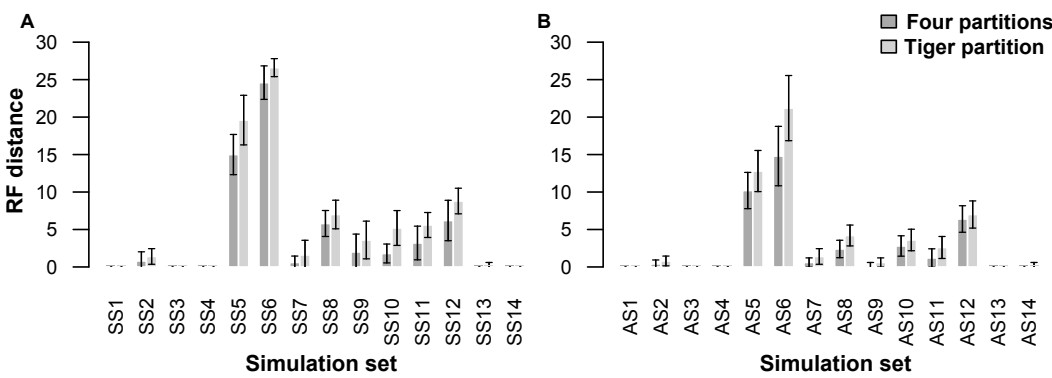

**Figure 1** Graphs showing Robinson-Foulds distances between the initial tree ((A) symmetrical tree, (B) asymmetrical tree) used for simulations and the trees inferred from simulated data. Each pair of bar plots corresponds to the mean values obtained from a different set of parameter values, when using four partitions (dark grey) or TIGER partitions (light grey) from ten replicates for each set of simulation conditions.

each type of trees (AS1, AS3, AS4, and AS13; SS1, SS3, SS4, and SS14) the two partitioning strategies systematically recovered the correct tree topology. Overall, more incorrect trees were inferred for the SS datasets than the AS datasets and this was consistent across the two partitioning strategies. The two partitioning strategies behave identically when the difficulty of the dataset increased. Datasets simulated with extremely short internal branches of 0.001 (S5, S6) or with an extreme amount of among-site rate variation together with relatively short branches (S8, S10, S11, S12), resulted in almost all cases in incorrect trees for both the 4-part and TIG analyses. The most extreme departures from the original trees occurred for simulations on trees with extremely short internal branches of 0.001. Under such conditions, even partitioning the dataset according to the initial simulation conditions failed to recover the correct topology, barely performing better than TIGER partitioning (RFD 4-part partitioning: AS5 mean $10.2 \pm 3.8$, AS6 mean $14.8 \pm 6.3$, SS5 mean $15.0 \pm 4.2$, SS6 mean $24.6 \pm 3.5$; TIGER partitioning: AS5 mean $12.8 \pm 4.3$, AS6 mean $21.2 \pm 6.9$, SS5 mean $19.6 \pm 5.2$, SS6 mean $26.6 \pm 1.9$). (Table 4, Fig. 1).

Our partitioning method performed equally well even when 25% of the data were missing for three taxa in the alignment (Table 5, Fig. 2). In the AS1 set of simulations, both 4-part and TIG strategies recovered the true tree in all cases, whereas the TIG partitioning performed slightly worse than the four-part in the AS2, SS1, and SS2 simulation sets, as was expected, but RFDs were quite low (RFD 4-part partitioning: AS1 mean $0 \pm 0$, AS2 mean $0.8 \pm 1.0$, SS1 mean $0 \pm 0$, SS2 mean $0.6 \pm 1.3$; TIGER partitioning: AS1 mean $0 \pm 0$, AS2 mean $1.0 \pm 1.4$, SS1 mean $0.4 \pm 0.8$, SS2 mean $1.0 \pm 1.1$) (Table 5, Fig. 2).

## Phylogenetic empirical datasets

The eight empirical datasets analysed covered a range of taxonomic ranks within Lepidoptera, from genus level (*Morpho* and *Calisto*), subtribes (Arctiina and Coenonymphina), two small to medium-sized families (Choreutidae and Pieridae, with about 400 and 1,100 species, respectively), to two very large families (Geometridae
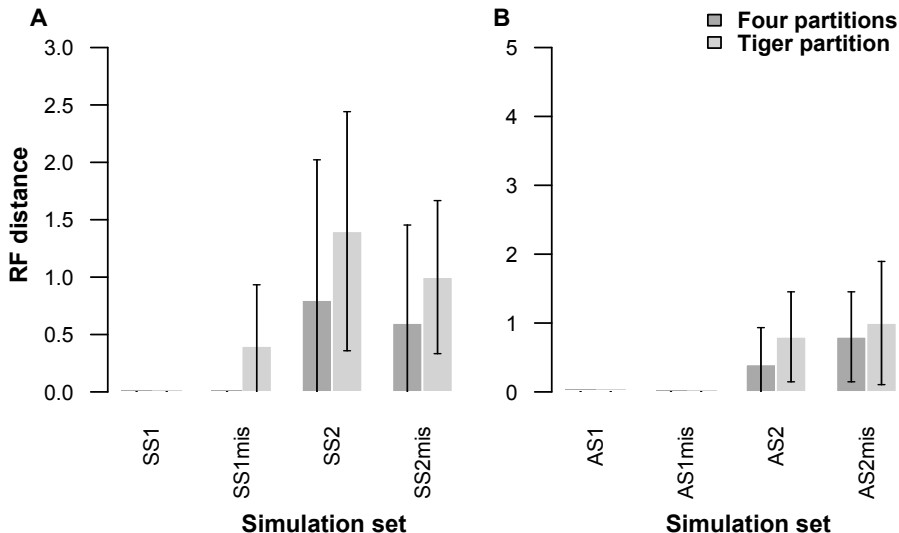

**Figure 2** **Graphs showing Robinson-Foulds distances between the initial tree used for simulations ((A) symmetrical tree, (B) asymmetrical tree) and the trees inferred from simulated data with and without missing data.** Each pair of bar plots corresponds to the mean values obtained from a different set of parameter values, when using four partitions (dark grey) or TIGER partitions (light grey) from ten replicates for each set of simulation conditions. SS1mis, SS2mis, AS1mis, and AS2mis correspond to the four cases where the datasets contained missing data.

**Table 5** **Tree topology comparisons for simulated datasets with missing data.** Shown are means and standard deviation for Robinson-Foulds distances between the inferred trees from simulated data compared with the true tree using the 4-partitioning strategy and the TIGER strategy.

|  | AS1 | | AS2 | | SS1 | | SS2 | |
|---|---|---|---|---|---|---|---|---|
|  | **mean** | **S.D.** | **mean** | **S.D.** | **mean** | **S.D.** | **mean** | **S.D.** |
| 4-part | 0.0 | 0.0 | 0.8 | 1.0 | 0.0 | 0.0 | 0.6 | 1.3 |
| TIGER | 0.0 | 0.0 | 1.0 | 1.4 | 0.4 | 0.8 | 1.0 | 1.1 |

and Noctuidae, with over 23,000 and 11,000 species, respectively) (*van Nieukerken et al., 2011*). The amount of missing data was quite variable. The most complete dataset, Coenonymphina, had more than 90% of sites with less than 20% of missing data, while the least complete dataset, Arctiina, had only 21% of sites with less than 20% of missing data (Table S1).

TIGER partitioning resulted in a different number of partitions for each dataset, with Geometridae and Pieridae being split into many more partitions than the other datasets (Table S2). For example, at $d$ equalling 4.5, *Morpho*, the dataset with fewest taxa, was split into only seven partitions, Pieridae into 20, Geometridae into 24, while all the other datasets ranged 10–14 in their number of partitions.

In all cases partitioning by gene region was clearly the worst way to subdivide the data, as determined by BIC scores, and applying the greedy search made little improvement (Fig. 3, Table S3). In all datasets, the model with partitioning using TIGER and RatePartitions was the best fit for the data. However, in two datasets (Geometridae and Pieridae),

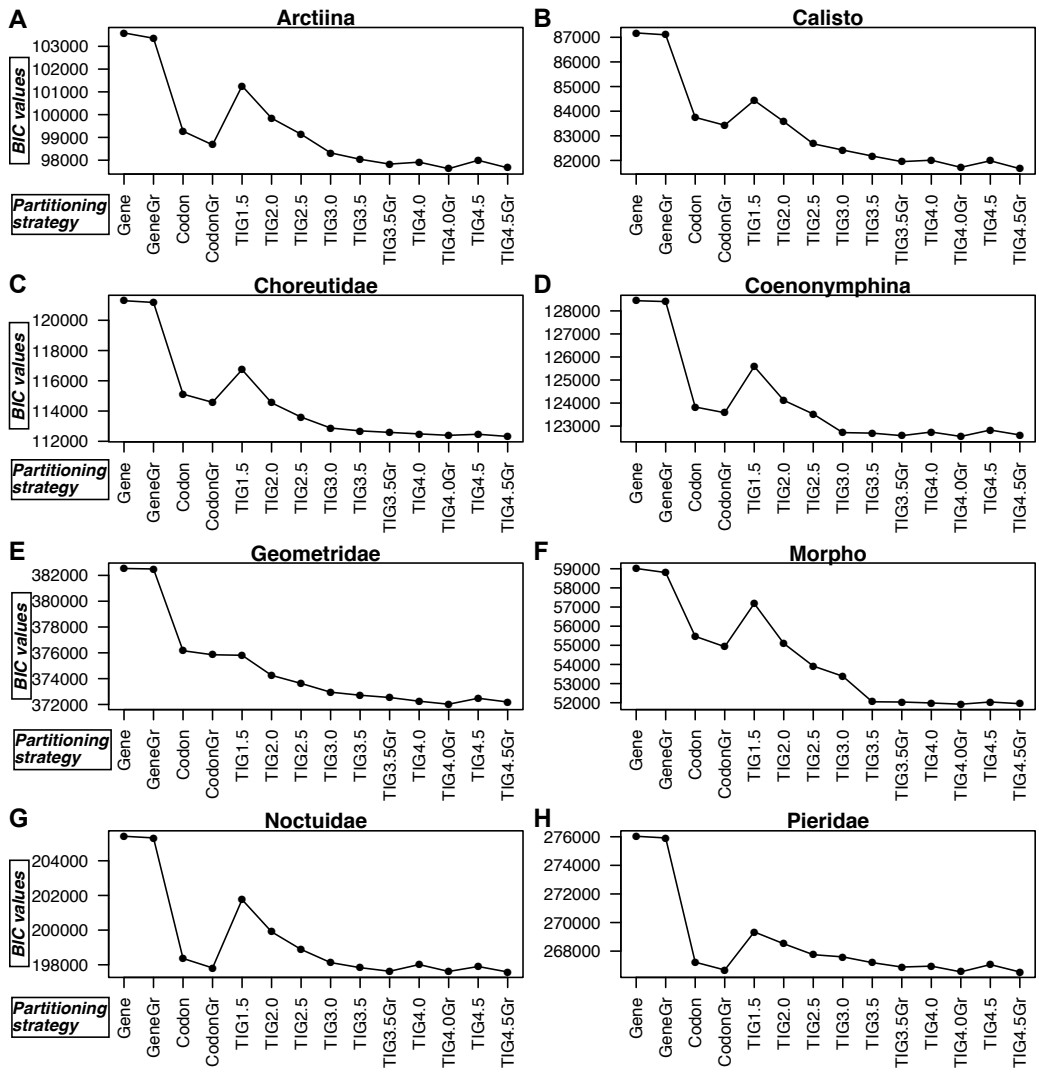

**Figure 3** **A comparison of BIC values for the 14 partitioning strategies tested in all eight datasets.** (A) Arctiina, (B) *Calisto*, (C) Choreutidae, (D) Coenonymphina, (E) Geometridae, (F) *Morpho*, (G) Noctuidae, and (H) Pieridae. The partitioning strategies are plotted on the horizontal axis, and the BIC values are plotted on the vertical axis. The lower the BIC value, the better the partitioning strategy. Gene refers to partitioning by gene fragment, *Codon* to partitioning by codon position, *TIG* to partitioning by relative evolutionary rate as estimated with the program TIGER with different values for the *d*, division factor in the RatePartitions script, and *Gr* refers to the strategies that included a greedy search. See text and Table 3 for more details.

partitioning by gene and codon position with a greedy search came close to the best TIGER strategy, although the BIC scores were still significantly better for the TIGER strategy (Table S3). In all datasets, the improvement in the BIC score from TIG1.5 to TIG3.0 was quite steep, but further differences between TIG3.5, TIG4.0, and TIG4.5, with and without greedy search were relatively small, although the analyses with the greedy search always received a significantly better BIC score. TIG4.5Gr was the best strategy in *Calisto*,

Choreutidae, Noctuidae, and Pieridae, whereas TIG4.0Gr was the best strategy in Arctiina, Coenonymphina, Geometridae, and *Morpho* (Fig. 3, Table S3).

A comparison of the trees resulting from the RAxML analyses of the best TIGER partitioning strategy with those from the CodonGr analyses revealed that the different datasets responded differently to the two partitioning strategies. For two datasets, Choreutidae and Pieridae, RFDs showed very low values, ca. 5% of internal branches were different between the two partitioning strategies (Table S4). However, most datasets showed great variation between the partitioning strategies, with >20% of internal branches differing. The differing branches tended to be very poorly supported regardless of partitioning strategy (Fig. S1).

An examination of the plots of the relative evolutionary rates estimated by TIGER-rates for each gene fragment and codon position reveals differences among gene fragments, as well as sites belonging to the same codon position in the same gene fragment (Fig. 4, S2). As expected, in general, first and second codon positions receive a much higher rate (i.e., implying slower change) than third codon positions, but in some genes there is a large proportion of third codon positions that also receive a rate of one, e.g., in the *Morpho* dataset for CAD, EF-1 $\alpha$, and RpS5 (Fig. S2). Conversely, there are genes that tend to have some fast-changing first and second codon positions, which then receive a relatively low rate. This is usually the case in COI, the mitochondrial gene, but also in several nuclear genes (wingless in all datasets, but also CAD, MDH, and RpS5 in some of the datasets; Fig. 4, S2).

### Phylogenomic empirical dataset

When we applied TIGER-rates and RatePartitions on the phylogenomic dataset from *Meiklejohn et al. (2016)* we recovered an almost identical tree except for the following three differences (Fig. S3). The relationships between *P. cristatus*, *G. gallus* and *A. rufa* shifted from the well supported (*P. cristatus*, (*G. gallus*, *A. rufa*)) in *Meiklejohn et al. (2016)* to poorly supported (*A. rufa*, (*G. gallus*, *P. cristatus*)). Two other nodes in *Meiklejohn et al. (2016)* had low bootstrap values (63 and 55). In our analyses using IQTree and the *Meiklejohn et al. (2016)* partitioning strategy, the bootstrap values for these same two nodes were much higher (93 and 82 respectively), and with the TIGER partitioning the bootstrap for the first node even further increased to 97, while for the second node it decreased to 74. Finally, we observed a great decrease of branch length at the base of the tree when using TIGER.

## DISCUSSION

Many studies have shown that partitioning of DNA sequence data for phylogenetic analysis is important because it affects the resulting tree topology, branch support, as well as branch lengths (see *Kainer & Lanfear, 2015* and references therein). A common approach is to define partitions *a priori* based on some feature(s) of the DNA sequences such as genes, codon positions, stems, loops, introns, exons, etc., but this can be problematic because the properties of the sequence data are not fully known to the user to begin with. To avoid *a priori* partitioning, we developed a method of partitioning based on relative evolutionary rates of sites in an alignment.

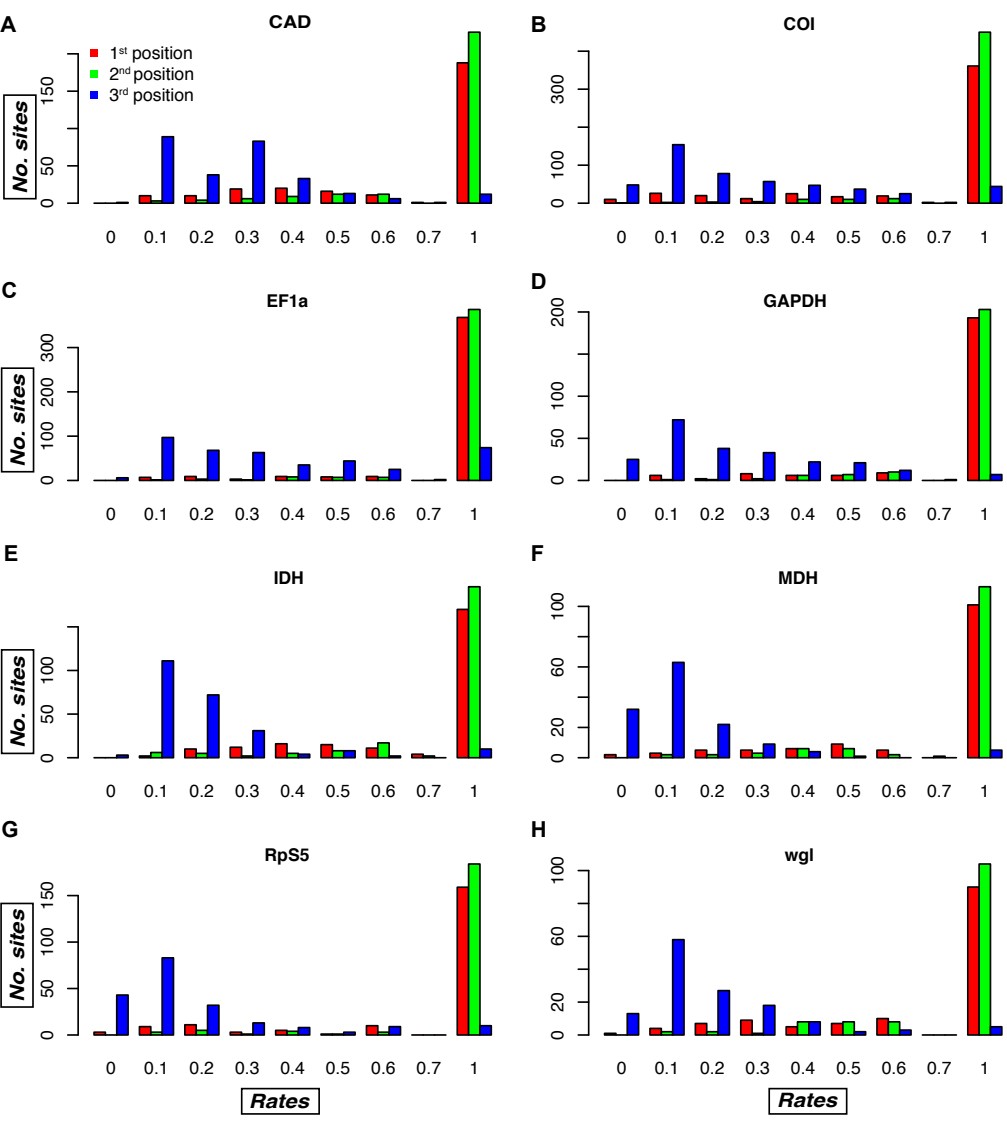

**Figure 4 Relative evolutionary rate estimates for codon positions in the Noctuidae dataset.** Plots are showing the assigned TIGER relative evolutionary rates for codon positions of each of the eight genes in the Noctuidae dataset: (A) CAD, (B) COI, (C) EF-1a, (D) GAPDH, (E) IDH, (F) MDH, (G) RpS5, and (H) wgl. TIGER rates are shown on the horizontal axis, and the number of codon positions that were assigned the rate between 0.0–0.1, 0.1–0.2, etc. is shown on the vertical axis. The lower the number, the higher the rate of evolution, with rate of 1 being assigned to invariable sites in the alignment. As expected, most of the first and second codon positions received the rate of 1, but there are exceptions, with some first and/or second codon positions receiving a relatively low rate (especially in e.g. COI, and wgl). Likewise, most of the third codon positions received lower rates, but in some genes (e.g. EF-1a), the number of third positions that received the TIGER rate of 1 is relatively high. Such plots for the other seven datasets are in Fig. S2.

Our simulations on a wide range of parameters showed that partitioning the simulated datasets with TIGER-rates and RatePartitions behaved identically to partitioning according to the initial conditions (four partitions). When the parameter conditions were more difficult, both partitioning strategies failed to recover the correct topology, although, as

expected, TIGER performed slightly worse than the four-partition strategy. Hence, our simulation tests did not highlight any misbehaviour of our method, even under extreme parameter conditions.

Comparing the fit of different partitioning strategies to empirical datasets, we demonstrated that our proposed method using TIGER and RatePartitions outperformed in all datasets that we tested other commonly used partitioning strategies, such as partitioning by gene and codon position.

A possible explanation for why TIGER partitioning performed better than partitioning by codon position is that there are significant differences among sites belonging to the same codon position of the same gene in their relative evolutionary rate (Fig. 4, S2), and this leads to high heterogeneity in the data when they are simply grouped by codon position. Since our method groups sites based on the pattern present in the alignment, the models of molecular evolution have to account for less variation within each partition.

In all of our analyses, partitioning by gene was much worse than the other strategies. A protein-coding gene, with its first, second, and third codon positions, each of which evolve differently, is highly heterogeneous, and applying the same model to such a sequence most likely leads to an underparameterized model. It has been demonstrated that underpartitioning can result in in a more severe error in most datasets than overpartitioning (*Brown & Lemmon, 2007*; *Ward et al., 2010*; *Kainer & Lanfear, 2015*), and our recommendation is to take this into account when devising a partitioning strategy.

Comparisons of phylogenetic trees resulting from analysing differently partitioned empirical datasets again demonstrate that partitioning does affect the topology, nodal support, and branch lengths, as has already been demonstrated in other studies (e.g., *Rota, 2011*; *Rota & Wahlberg, 2012*). Most of these differences occur in areas with relatively poor support. The comparisons between partitioning strategies using TIGER or by gene and codon position served mainly to highlight the parts of the tree that are poorly supported by the data at hand, a result that is in line with our simulation study. We show that phylogenomic datasets can be partitioned by relative rates into subsets. This is particularly of concern when dealing with huge datasets, where partitioning by gene and codon position is not feasible. Although there were some differences between our results and the analysis published in the *Meiklejohn et al. (2016)* study, the trees are largely similar in topology and branch supports.

Additionally, our partitioning method has been applied in analyses of several other lepidopteran datasets: (1) the subfamily Acronictinae (Noctuidae) (*Rota et al., 2016*) analyzed in MrBayes (*Ronquist et al., 2012*) and RAxML (*Stamatakis, 2014*); (2) an expanded dataset for the family Choreutidae (*Rota, Pena & Miller, 2016*) in MrBayes, RAxML, and BEAST (*Drummond et al., 2012*); (3) the family Hesperiidae (skippers) (*Sahoo et al., 2017*) in BEAST; (4) the family Riodinidae (metalmark butterflies) (*Seraphim et al., 2018*); (5) butterfly subfamily Limenitidinae (*Dhungel & Wahlberg, 2018*); and (6) for inferring relationships among Ditrysian superfamilies and families using molecular and morphological characters (*Heikkilä et al., 2015*) in RAxML. In the *Heikkilä et al.* study (*2015*), in addition to applying our partitioning method, the authors also explored the effect of exclusion of fastest evolving characters from the analyses. They found that

phylogenetic signal was lost especially when the fastest evolving morphological characters were excluded, and that branch support was lowered with the exclusion of fastest evolving molecular characters, which also resulted in a spurious placement of some groups, and therefore is not at all recommended (see *Simmons & Gatesy, 2016* for a detailed exploration of this topic).

The TIGER method is entirely based on the alignment—not on trees or some features of the data deemed important by the user. It can be applied to any kind of categorical data (nucleotides, amino acids, morphological characters), to protein-coding genes, RNA, introns, exons, as well as UCEs. It can be especially useful for sequences derived from introns or UCEs, where *a priori* partitioning is difficult, as one does not need to provide user-defined partitions. However, our method has not yet been tested in timing of divergence analyses so we recommend careful examination of results in such cases.

An issue we would like to stress with our approach, however, is that it should only be applied to studies where concatenation of data is justified, i.e., where gene tree/species tree problems are minimized. This is because our approach of partitioning by specific properties of each character removes any connections between characters belonging to the same gene region. This reshuffling of characters based on relative rates of evolution does have a biological basis to it (sites evolving at a similar rate are modelled together), but at the risk of losing other biologically relevant information (such as differential evolutionary histories of gene regions). We do feel that for studies looking at deeper relationships, such as among genera, tribes, families, and orders, our approach is very useful and overcomes problems of overpartitioning for large multigene datasets that might be partitioned by codon position, as well as underpartitioning when users might be inclined to analyse their data unpartitioned because they are uncertain of how to partition *a priori*.

## CONCLUSIONS

Here we present a way of partitioning data based on relative rates of evolution as calculated by TIGER-rates. Analyses of simulated data over a wide range of parameter space did not reveal any misbehaviour of the method. Furthermore, we demonstrate that this approach works better than the traditional approaches based on the model BIC scores in eight empirical phylogenetic datasets and that its performance was similar in a phylogenomic dataset. Further utility of TIGER calculated rates and RatePartitions needs to be ascertained on other datasets. The program could certainly be used on amino acid (or any other categorical) data in the same way as done here for nucleotides. We have tested the method using one phylogenomic dataset (*Meiklejohn et al., 2016*), but it would be desirable to test this method with additional phylogenomic datasets that contain sequences from hundreds or thousands of genes. It is also important to recognize that the relatively simple stochastic models used for phylogenetic analyses represent approximations to much more complex processes (*Wilke, 2012*). There are likely to be some datasets where available models are simply inadequate (*Reddy et al., 2017*). However, practicing systematists should always endeavour to find the best-fitting model that can be implemented in analytical programs that are both available and practical. The method we present is an important step in that direction.

## ACKNOWLEDGEMENTS

We are grateful to Edward L. Braun (the editor), Claus Wilke, Robert Lanfear, and Mark Simmons (the reviewers) for very careful and helpful reviews.

### Funding

This work was supported by the Kone Foundation funding to Jadranka Rota and Tobias Malm, and Academy of Finland and the Swedish Research Council funding to Niklas Wahlberg. The funders had no role in study design, data collection and analysis, decision to publish, or preparation of the manuscript.

### Grant Disclosures

The following grant information was disclosed by the authors:
Kone Foundation.
Academy of Finland.
Swedish Research Council.

### Competing Interests

Carlos Peña is employed by HipLead, San Francisco, CA, United States of America.

### Author Contributions

- Jadranka Rota conceived and designed the experiments, performed the experiments, analyzed the data, contributed reagents/materials/analysis tools, prepared figures and/or tables, authored or reviewed drafts of the paper, approved the final draft.
- Tobias Malm conceived and designed the experiments, performed the experiments, analyzed the data, contributed reagents/materials/analysis tools, authored or reviewed drafts of the paper, approved the final draft.
- Nicolas Chazot performed the experiments, analyzed the data, contributed reagents/materials/analysis tools, prepared figures and/or tables, authored or reviewed drafts of the paper, approved the final draft.
- Carlos Peña performed the experiments, analyzed the data, contributed reagents/-materials/analysis tools, authored or reviewed drafts of the paper, approved the final draft.
- Niklas Wahlberg conceived and designed the experiments, performed the experiments, analyzed the data, prepared figures and/or tables, authored or reviewed drafts of the paper, approved the final draft.

### Data Availability

The raw data have been supplied as Supplemental Files, as well as on Zenodo, and the python script has been deposited at GitHub.

Zenodo: https://zenodo.org/record/1252828#.Ww4M81Mvxao, https://zenodo.org/record/1251684#.Ww4NAlMvxao

GitHub: https://github.com/jadrankarota/RatePartitions.

## Supplemental Information

Supplemental information for this article can be found online at http://dx.doi.org/10.7717/peerj.5498#supplemental-information.

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
