# Peer review of "A simple method for data partitioning based on relative evolutionary rates"

_PeerJ, doi:10.7717/peerj.5498_

## Round 0.1 · original submission · Major Revisions

Your manuscript has now been reviewed by three experts in the field. All of these reviewers recommended major revisions and I concur with their evaluation. I found all of their comments to be valuable, although it is perhaps unfortunate that all three had slightly different concerns. I think that trying to address all of them might place an unfair burden on a manuscript that has (in my opinion) fairly clear value. For that reason, I am going to give some guidance regarding what I believe to be the “big picture” aspects of the reviews. I hope I am able to strike a balance between the amount of work I am asking you to do and the value that work will add to the manuscript.

There are also a number of “minor comments” in the reviews and I hope the authors simply address those comments to the best of your ability.

All three reviews are public reviews, so I will identify the reviewers in my comments.

Now to the bigger issues:

1. The program needs to be placed in a permanent repository like github or zenodo (or dryad or figshare). It would be good to post the datasets too (zenodo, figshare, or dryad would be good for this, though I believe you would have to pay for Dryad). Since the datasets are published they may already be available. If they are already available provide the link somewhere (perhaps in table 1). If they are not available, please make them available.

2. Although the authors are certainly correct when they state that “…true phylogenies in all of these cases are unknown, we can only select the best 
partitioning strategy using statistical model evaluation metrics, such as e.g. BIC.” I don't think that is a good rationale to avoid running a tree search with bootstrapping and showing whether their partitioning method affects topology and/or support. They could do this search in either RAxML or IQ-TREE. However, if they use IQ-TREE they should acknowledge that the ultrafast bootstrap (Minh et al. 2013) has a different scale from the standard bootstrap (or the RAxML rapid bootstrap).

3. I concur with Dr. Lanfear that this paper needs a simulation to show that the partitioning method recovers reasonable trees. Of course, setting up appropriate and useful simulations can be challenging, and I do not think that it is possible to answer every question with any simulation. One could argue that Dr. Lanfear’s suggested simulation is unfair to your partitioning method (indeed, to any partitioning method) since the simulation he described has a single partition. But the simulation would show that nothing in your partitioning method distorts the tree, at least given that the simulation conditions. Showing that is important.

Given Dr. Lanfear’s offer to provide code I would reach out to him and conduct the suggested simulation. I think that four taxon simulations would be pointless; it seems to me that estimating site rates with TIGER would be difficult for four taxa and few researchers analyze four-taxon datasets in this day and age. I’d shoot for at least 10 taxa. I’m going to suggest another simulation with 16 taxa and perhaps that is the number to go for here.

I would like to add one simulation idea that would potentially address Dr. Wilke’s concerns (more about that below, after I describe the mechanics).

Simulate 4 partitions, each of 1 kb. Conduct the simulations using F81+G but vary the base composition (e.g., partition one use %A=%T=40; %G=%C=10; partition two use %A=%T=30; %G=%C=20; partition three use %A=%T=20; %G=%C=30; partition two use %A=%T=10; %G=%C=40). Use the same alpha parameter for each partition. Concatenate the partitions and then proceed with the TIGER partitioning. Compare to partitioning using based on the partitions you simulated (i.e., the four 1kb partitions)

I just suggested some values you might want to try. But you should probably experiment. If it would be easy to do this using Dr. Lanfear’s code to simulate tree topologies then do that. If it is challenging to use Dr. Lanfear’s code to do these simularions you could try a maximally symmetric and a maximally asymmetric tree and do the simulations in a program like Indelible (you do not have to do the realism of simulating indels; in fact, you should avoid indels so you can stay focused on the model specification/misspecification issue). These trees would be:

(out,(((A,B),(C,D)),((E,F),(G,H))),(((I,J),(K,L)),((M,N),(O,P))));
(out,P,(O,(N,(M,(L,(K,(J,(I,(H,(G,(F,(E,(D,(C,(B,A)))))))))))))));

(note that these are actually 17-taxon trees because I’ve included an outgroup). Just set the internal branches to some short branch length (try 0.01, 0.005, and 0.001) and the terminal branches to 0.1 (obviously, the asymmetric tree will be non clocklike). Not that Indelible will simulate alignments given multiple models, so you don’t even need to concatenate.

Minimally, I recommend trying the basic simulations outlined above (i.e., the JC+G simulations and the F81+G mixed simulations). But I would love to see the results of tweaking it to make the partitions evolve at different overall rates (e.g., multiply all branch lengths for partition 2 by 2, partition 3 by 3, and partition 4 by 4 – again this is just an example, there is nothing magic about the suggested values). Your partitioning method might actually perform better in this “different rates for different partitions” part of parameter space than in the part of parameter space where all partitions have equal rates.

Note that the F81+G simulations should reduced to JC+G when all partitions have 25% each nucleotide.

The point behind the F81+G simulation is related to Dr. Wilke’s comment that you should think “…more carefully about why different sites evolve at different rates.” I agree with that comment. In my own work (Reddy et al. 2017) I pointed out the following: “However, the complexity of the factors that determine rates of amino acid substitution (for recent reviews, see Chi and Liberles 2016; Echave et al. 2016) suggests this is unlikely to be the case. For example, constraints that result in slower accumulation of substitutions in coding regions could drive site-specific biases in nucleotide frequencies due to correlation between physicochemical properties of amino acids and the structure of the genetic code (e.g., Supplementary Fig. S4, available on Dryad).” The point is that the conservation of amino acid physicochemical characteristics combined with the structure of the genetic code might drive the conservation of different equilibrium amino acid frequencies at different sites (e.g., a buried site in a globular protein might be free to vary. Obviously, protein evolution is complex, but this might be a first step.

I’ve thought quite a bit these simulations and I’ve tried to balance the amount of work you’ll need to do with the value of what we’ll learn. I think I’ve hit a reasonable balance, but I’m certainly willing to discuss the details. I think your method has potential, but I think the community should have at least some simulations as a starting point to evaluate it.

You can use either RAxML or IQ-TREE to analyze the simulated data, though I will say that IQ-TREE is fast, easy to use, and appears to perform well (Zhou et al. 2017).

4. I think Dr. Simmons’ comment about the bias in TIGER with respect to the amount of possible synapomorphy is important to address. I would assess the amount of possible synapomorphy for each partition in each partition. This can be done in TNT using the following:

log `_minmax.log;

minmax;

log/;

log `_opt.log;

Alternatively, it should be possible to do this in PAUP*, though I believe that doing so would be a bit harder for a site-by-site analysis of the amount of possible synapomorphy. Please contact me if you have concerns about the mechanics of doing this.

5. Regarding Dr. Simmons’ question “given that these models invoke different numbers of partitions and hence parameters, how did they control for this effect on their Y-axis?” my reading of the paper is that your y-axis is BIC, not likelihood. This would control for the number of free parameters. However, the axis is not labeled in the figure. Doing so would clarify this and address the reviewers’ question. Adding a justification for using BIC rather than AIC/AICc is also important.

6. Both Dr. Simmons and Dr. Lanfear bring up the basic question of why the authors view a partition with only invariant sites as a source biased likelihood values. As I see it, an all invariant partition is not intrinsically biased as long as the sites in that partition are genuinely invariant in the unknown (and unknowable) true model. The unknowability of “true models” in phylogenetics is, of course, the fundamental philosophical issue associated with parametric methods in phylogenetics. The philosophical discussion is outside the scope of the present work, so I’ll move forward with the assumption that the authors’ method is a way to identify a good approximating model.

In that context, the potential bias enters with the fact that one expects some proportion of sites to be invariant even if those sites can vary. This will be true unless the treelength is very long (in expectation I believe there will be some invariant sites as long as the treelength is finite). However, this raises an issue. If you take the sites that appear invariant but are potentially variable and put them in an invariant partition this causes a bias in all partitions, not just the invariant sites partition. Perhaps this is the issue the authors are alluding to. Regardless, this should be discussed in a bit more detail.


7. One additional issue is that the authors state “It can be applied to any kind of categorical data (nucleotides, amino acids, 
morphological characters), to protein-coding genes, RNA, introns, exons, as well as ultra-conserved elements (UCEs). It can be especially useful for sequences derived from introns or UCEs, where a priori partitioning is difficult, as one does not need to provide user-defined 
partitions.” This is exciting because introns and UCEs do not have obvious partition information associated (although, as Dr. Wilke notes, protein evolution is more complex than 1st, 2nd, and 3rd codon positions). However, the authors do not test any UCE or intron data. I think it would behoove them to do so, especially since “UCEs” is one of their keywords (I note that “phylogenomics” is also a keyword and I don’t consider any of their test datasets phylogenomic in scale).

I recommend the authors include at least one UCE test dataset. I recommend a dataset from my own lab (Meiklejohn et al. 2016), which has the benefit of including a manageable number of taxa for analysis and data matrix comprising UCEs, introns, mitochondrial coding regions, and a mitochondrial rRNA. This mixture of data could be interesting for your partitioning methods and the Meiklejohn et al. (2016) paper is the only one (to my knowledge) that includes a fully sampled matrix with all of those types of data. The authors are free to include any other UCE data that they want (especially if they are aware of any other mixed data matrix comparable to Meiklejohn et al. 2016). Or they could simply use the Meiklejohn et al. (2016); the only disadvantage of those data is the fact that analyses are equivocal for one specific relationship. However, if the authors were willing to conduct their partitioning and then conduct a tree search it would be very interesting if their partitioning method improves support for any specific resolution!


In summary, I like this paper. I think this method has promise. But I don’t think it is sufficient to just show that BIC scores are improved on a limited set of datasets. I hope that I have balanced the amount of extra work necessary to show readers that the method does will indeed be useful.

References:

Chi P.B.; Liberles D.A. 2016. Selection on protein structure, interaction, and sequence. Protein Sci. 25:1168–1178.

Echave et al. 2016 reference available below in Dr. Wilke’s review.

Meiklejohn, Kelly A.; Brant C. Faircloth; Travis C. Glenn; Rebecca T. Kimball; Edward L. Braun. Analysis of a Rapid Evolutionary Radiation Using Ultraconserved Elements: Evidence for a Bias in Some Multispecies Coalescent Methods. Systematic Biology, Volume 65, Issue 4, 1 July 2016, Pages 612–627, https://doi.org/10.1093/sysbio/syw014

Minh, Bui Quang; Minh Anh Thi Nguyen; Arndt von Haeseler. Ultrafast Approximation for Phylogenetic Bootstrap, Molecular Biology and Evolution, Volume 30, Issue 5, 1 May 2013, Pages 1188–1195, https://doi.org/10.1093/molbev/mst024

Reddy, Sushma; Rebecca T. Kimball; Akanksha Pandey; Peter A. Hosner; Michael J. Braun; Shannon J. Hackett; Kin-Lan Han; John Harshman; Christopher J. Huddleston; Sarah Kingston; Ben D. Marks; Kathleen J. Miglia; William S. Moore; Frederick H. Sheldon; Christopher C. Witt; Tamaki Yuri; Edward L. Braun. Why Do Phylogenomic Data Sets Yield Conflicting Trees? Data Type Influences the Avian Tree of Life more than Taxon Sampling, Systematic Biology, Volume 66, Issue 5, 1 September 2017, Pages 857–879, https://doi.org/10.1093/sysbio/syx041

Zhou, Xiaofan; Xingxing Shen; Chris Todd Hittinger; Antonis Rokas. Evaluating fast maximum likelihood-based phylogenetic programs using empirical phylogenomic data sets. Molecular Biology and Evolution, msx302, https://doi.org/10.1093/molbev/msx302 (advance access, Published: 21 November 2017)

·

Basic reporting

see below

Experimental design

see below

Validity of the findings

see below

Additional comments

Rota and colleagues present a generally well written manuscript that formalizes, tests, and makes publicly available a novel method that they have used in some recently published papers. I think the manuscript merits publication in PeerJ, but only after the authors have rigorously addressed my six sets of comments below.

-- Mark P. Simmons, 16 November 2017

Major comments:

I’m glad to see that the authors appropriately cite John Gatesy’s and my 2016 paper criticizing TIGER (and OV). But so far as I can tell they have not addressed the implications of our study on their favored method of using TIGER to partition characters by rate. Even when all characters are sampled, isn’t the approach still biased in favoring assignment of informative characters with highly asymmetrical partitions to the slowest rates? The authors can test for this using Farris’ (1989) amount of possible synapomorphy. Also, do the trees generated using their partition-delimitation method have more cherries than the trees using the other four partitioning methods they applied? See Simmons and Gatesy (2016) for additional details. I think the authors need to address this bias before recommending their methodology for general use.

“However, the k-means algorithm has been found to place all invariable characters into one partition (Baca et al., 2017), which leads to biased likelihood values." The point is to partition characters by rate, and the simplest explanation for invariable characters is that they have not evolved at all, and hence should be assigned to the lowest rate partition. State explicitly why you believe this approach is problematic. The authors cite biased likelihood values, but why does this approach lead to biased likelihood values and why does the approach fail based on theory rather than in an operational context?

The main results of the paper are presented in Table 1. First, why wasn't TIGER, without the authors' method tested? Second, given that these models invoke different numbers of partitions and hence parameters, how did they control for this effect on their Y-axis? So far as I can tell there is no control and so it's perfectly reasonable to expect more partitions to lead to greater likelihoods.

Minor comments:

“We do not carry out phylogenetic analyses and compare
resulting topologies because it has been previously established that partitioning does affect
topology, branch support, and branch lengths (see Kainer & Lanfear, 2015 and references
therein), and since true phylogenies in all of these cases are unknown, we can only select the best
partitioning strategy using statistical model evaluation metrics, such as e.g. BIC." Break that up into two separate sentences.

Where is the description of how tree searches were performed? I don't see this information in the methods.

“BIC was chosen as a statistical model evaluation metric because it has been shown to perform well in model selection for phylogenetic analysis (Abdo et al., 2005)." The same can certainly be said for the AIC and the LRT. State explicitly why you chose to use the BIC rather than these alternatives.

·

Basic reporting

Two minor comments:

1. The figures need axis labels at all axes.

2. l. 104-106 "One of these properties is that with standard DNA sequence data of protein-coding genes, one to two thirds of the data consist of invariable characters." This seems to depend on the specific data set analyzed. Maybe write "typically consist of".

Experimental design

One comment:

An improved BIC score does not in any way guarantee that the inference result is closer to the truth. This is particularly the case when the model is misspecified, which is essentially always the case when working with real (as opposed to simulated) data.

I think that, at a minimum, the authors should also perform a test on simulated data and show how their method affects actual quantities of interest, such as the distance between the inferred and the true tree.

Validity of the findings

Two comment:

1. As mentioned under "Experimental design", improvement in BIC does not guarantee that the inferred result is actually closer to the truth. This needs to be discussed, and the method needs to be tested with some simulations where the ground truth is known.

2. I would like to see the authors think more carefully about why different sites evolve at different rates. It's not clear to me that all sites that evolve at the same rate can be thrown together into one group. For example, second codon positions may evolve rapidly if the corresponding amino acid lies on the surface of the folded protein [1], and third codon positions may evolve slowly due to various causes of codon bias [2]. Is there any good reason to believe that a fast-evolving second position and a slow-evolving third position should be grouped together?

[1] J. Echave, S. J. Spielman, C. O. Wilke (2016). Causes of evolutionary rate variation among protein sites. Nature Rev. Genet. 17:109–121

[2] J. B. Plotkin, G. Kudla (2011). Synonymous but not the same: the causes and consequences of codon bias. Nature Rev. Genet. 12:32–42

·

Basic reporting

Fine

Experimental design

Needs simulations and code tests, see below.

Validity of the findings

Depends on simulations and comparisons with related methods (free-rates models). See below.

Additional comments

This paper presents an approach to automatically partition datasets based on TIGER rates.

I am intrigued by the approach, but I have a couple of concerns which mean that I remain to be convinced that the method is useful.

First, I think it’s vital that we see some simulations here. I say this because we have tried a range of very similar approaches to this, and in every case we have found insurmountable problems with the approach on suitably simulated data. The authors correctly point out that our previous attempt at a very similar approach (the k-means method) had some unforeseen properties. In developing that method we tried it on a range of real and simulated datasets, and never found any odd behaviour. As the authors (and we) have noted, this was premature: our method turns out to do some odd things.

What the authors couldn’t have known is that in light of the problems identified with our method, we tried a range of approaches that are very similar to that presented here. These included (but weren’t limited to) lumping together invariant sites and slow sites; removing invariant sites from the analysis and using a bias correction when inferring trees; and lumping every invariant site with the physically closest variable site. In all cases, the methods seemed to perform just fine on most datasets. But in all cases we found that they had inexplicable issues in some corner cases. Because we were unable to resolve (or properly understand) these issues, we have yet to publish these results.

The relevance of this for the current ms is simple: we need to see a good range of simulations in which the authors simulate appropriate datasets (more on that below), and test whether using their partitioning method one can recover the correct tree and branch lengths. Given the history of these methods, it’s absolutely crucial that this is done before the method is published and promoted. To be clear – I absolutely hope that this method works great. It is certainly a little different from what we proposed in print, or have tried since. But to be frank, I suspect that it will suffer the same limitations. And if it does, I don’t think it should be published or promoted for use unless those limitations can be clearly understood, and crucially predicted accurately for empirical datasets. It’s the latter limitation which led us to remove k-means from PartitionFinder. All of our evidence suggested that the odd behaviour was very very rare. But unless we can predict which empirical datasets will suffer from the odd behaviour, we cannot (and the authors should not, if simulations show their method to also be inadequate) suggest that this is a useful method.

The simplest simulation conditions for which I have been able to show that the k-means algorithm (and our various modifications of it, which are similar to the work presented here) suffered from odd innacuracies are as follows:

1. Simulate a phylogram with rate variation, with 4-20 taxa
2. Simulate a 4kb alignment under a JC+G model on that phylogram
3. Reconstruct the tree (using e.g. IQtree) with the true model (i.e. JC+G)
4. Perform the TIGER partitioning described here
5. Reconstruct the tree under the TIGER partitioning scheme

If one repeats this ~10 times for each set of simulation conditions (e.g. varying the number of taxa, the total substitution rate on the tree, and the alpha parameter of G), one has a baseline set of simulations where rate variation varies along (because of variation in G). Of course, one will also need to vary the parameter for rate splitting presented in the current method.

Reconstructing the tree with the true model tells one how accurate one can hope to be (on average across the 10 simulations). And one can then ask whether the TIGER rate partitioning is as accurate as analysing the tree with the true model, or less accurate. Of course, we would expect it to be a little less accurate. But what we observed with the k-means approach (especially on datasets with very few taxa) was that the reconstructed k-means trees often completely collapsed the internal branch.

I have already written all of the code for these simulations (except the TIGER parts, which would be simple to add) and would be happy to share this with the authors if that’s of use. I really, genuinely, hope that their method overcomes the limitation of the method we proposed. But in the circumstances I think it’s very important to check with some rigorous simulations. Crucially (and where we failed in our ms) these simulations have to include tests of the reconstruction of the tree itself.

Of note is that these simulations are extremely fast to do, even without access to HPC resources it would be possible to do 10s of thousands of simulations.

A few other comments:

Line 84: I don’t know of any evidence that the likelihood values are biased. The likelihood values make perfect sense – an invariant site can be perfectly explained with a zero-length tree, so gets a likelihood of 1. I think the issue comes in model-misspecification on the remaining subsets (those without invariant sites included). This is likely to be an issue for the current method too, because it removes invariant sites from all of the faster-evolving subsets. All the more reason for a rigorous set of simulations.

Lines 96-99. For all the reasons above, I think it is critically important to compare phylogenies. Both of the simulated data (see above) and the empirical datasets.

Lines 108-109: invariant partitions do contain information: on the distribution of rates across sites and on the base composition. These inform the model, which informs the tree.

Line 109: What is the evidence that it’s advisable to include some slowly-evolving sites in a partition of invariant sites?

Line 143-145: This sentence is unclear. In addition, no justification is provided for removing sites. Also, my (perhaps incorrect) understanding was the TIGER did not cope with missing data. Is this wrong? Can the authors explain how a TIGER rate is calculated when data is missing here.

General point: I think one should really compare this approach to the newly-implemented free-rate models in IQtree. These are mixture models which nominally do exactly what the authors are proposing, but without the headache of running TIGER (which is exceptionally slow for larger datasets) and the author’s script. This could easily be incorporated in the simulation pipeline, and also in the analysis of empirical datasets, simply by adding an analysis in which one ran IQtree with automated model selection. This is important, because it may be that the authors method outperforms (or is outperformed by) these methods. The advantage of the free-rate models is that they have a rigorous statistical underpinning, whereas the current method (and my own work on k-means) is much more of a hack to a solution. I have no problem with the hacked version, but the target audience for this method (empirical phylo folks) should be given a straightforward comparison of methods.

The Python script seems to work fine (I tried it). However, I have a few concerns given that the paper is proposing that this is a script that others should use. First, the script should ideally be placed in a repository (e.g. GitHub) where people can find it. Second, and more importantly, we need to know how the script was tested, and we need to see the tests. There should be a series of unit tests, and some end-to-end tests, which demonstrate that the script works as intended.

Yours,

Rob Lanfear

---

## Round 0.2 · Minor Revisions

I'd like to start by saying I am sorry. Between travel and feeling ill I was unexpectedly out of communication. So I am running late on responding to this. I hope you are willing to accept my apology.

Now on to the manuscript. I feel you have largely dealt with the criticisms of your initial submission. I just have a few things I would like to point out:

I debated how much you should deal with Dr. Wilke's comment that "it is critical to not pretend that actual genetic sequences in actual organisms are just random strings generated from some uniform stochastic process". Your manuscript describes a method and is not a treatise on the philosophy of phylogenetic estimation or an attempt to extend available models in a radically-different way. It is a nuts-and-bolts practical application that the community can use - and should consider using.

That said, I think it would be worth just acknowledging the issue. I would really like to see you modify your conclusions as follows:

Here we present a way of partitioning data based on relative rates of evolution as calculated by TIGER-rates. Analyses of simulated data over a wide range of parameter space did not reveal any misbehaviour of the method. Furthermore, we demonstrate that this approach works better than the traditional approaches based on the model BIC scores in eight empirical phylogenetic datasets and that its performance was similar in a phylogenomic dataset. Further utility of TIGER calculated rates and RatePartitions needs to be ascertained on other datasets. The program could certainly be used on amino acid (or any other categorical) data in the same way as done here for nucleotides. >>>WE HAVE TESTED THE METHOD USING ONE PHYLOGENOMIC DATASET (MEIKLEJOHN ET AL. 2016) BUT IT WOULD BE DESIRABLE TO TEST<<< this method >>>WITH ADDITIONAL<<< phylogenomic dataSETS THAT CONTAIN<<< sequences from hundreds or thousands of genes. >>>IT IS ALSO IMPORTANT TO RECOGNIZE THAT THE RELATIVELY SIMPLE STOCHASTIC MODELS USED FOR PHYLOGENETIC ANALYSES REPRESENT APPROXIMATIONS TO MUCH MORE COMPLEX PROCESSES (WILKE 2012). THERE ARE LIKELY TO BE SOME DATASETS WHERE AVAILABLE MODELS ARE SIMPLY INADEQUATE (REDDY ET AL. 2017). HOWEVER, PRACTICING SYSTEMATISTS SHOULD ALWAYS ENDEAVOR TO FIND THE BEST-FITTING MODEL THAT CAN BE IMPLEMENTED IN ANALYTICAL PROGRAMS THAT ARE BOTH AVAILABLE AND PRACTICAL. THE METHOD WE PRESENT REPRESENTS AN IMPORTANT STEP IN THAT DIRECTION.<<<

ADDITIONAL REFERENCES:

Wilke, C. O. (2012). Bringing molecules back into molecular evolution. PLoS computational biology, 8(6), e1002572.

Reddy, S., Kimball, R. T., Pandey, A., Hosner, P. A., Braun, M. J., Hackett, S. J., et al. (2017). Why do phylogenomic data sets yield conflicting trees? Data type influences the avian tree of life more than taxon sampling. Systematic biology, 66(5), 857-879.

Note that I have inserted my edits in all caps and indicated boundaries with >>> and <<<

I was a bit mixed about tooting my own horn by suggesting that you include the Reddy paper, but I think it dovetails nicely with Wilke's comment. I've always felt that there are two aspects to phylogenetics: the practical and the philosophical. To be practical we have to use models that get the job done. But if we're honest about philosophy we should acknowledge that the models aren't perfect. Your paper has its feet firmly planted in the practical camp. I hate to hold it up by debating the philosophical!

Two other minor glitches I'd like you to fix are:

lines 279-280:
either by partitioning according to TIGER-rates and RatePartitions or following Meiklejohn et al. (2016) partitioning strategy.
>>>DELETE THIS: in the original study (Meiklejohn et al., 2016).<<<

lines 458-460:
Although there were some differences between our results and the analysis published in the Meiklejohn et al. >>>(2016)<<< study, the trees are largely similar in topology and branch supports.

I really appreciate the excellent job you did playing with the Meiklejohn et al. (2016) data. These minor glitches always happen when adding to paper.

I hope you find these comments helpful. Do feel free to use as is or to tweak what I wrote. I'm just offering it in the spirit of being helpful! I think you've produced a nice method and I'd like to see the community start testing it broadly.

I don't think my comments will take long to address. I'm certain you can address them adequately and I hope to see the manuscript back soon. I'm looking forward to testing your method on my data!

·

Basic reporting

N/A

Experimental design

N/A

Validity of the findings

I don't feel the authors have adequately addressed my comments. I had two comments under this topic, one of which was "I would like to see the authors think more carefully about why different sites evolve at different rates" (reiterated by the handling editor). The authors essentially dismissed this point and did not make any relevant changes that I could see. I think this is insufficient.

I understand that this is a complex topic and there's only so much one can do in a paper, but at the same time I think it is critical to not pretend that actual genetic sequences in actual organisms are just random strings generated from some uniform stochastic process (see also: http://journals.plos.org/ploscompbiol/article?id=10.1371/journal.pcbi.1002572). There is real biophysics underlying the evolutionary divergence of genetic sequences, and at a minimum the authors need to think under what conditions their work may or may not violate known facts about biophysical constraints on molecular evolution. Pretending these issues can safely be ignored is insufficient.

What I would like to see is about a paragraph of a careful discussion of these issues.

Additional comments

N/A

---

## Round 0.3 · accepted · Accept

Thanks so much for your quick revision. I hope you didn't mind my pushing the specific edits, but I felt it was important to make the point that partitioning is important and useful but not a panacea. I think many practicing systematists understand this, but I feel it bears repeating.

Regardless, I think this works perfectly and I am happy to accept. Please accept my apology for taking about a week to get to your paper - I've been running quite behind this summer. I feel bad that it resulted in a delay for your final acceptance.

#